# Framework for Smart Cost Optimization of Material Logistics in Construction Road Projects

**Abdulkareem Alanazi** * , **Khalid Al-Gahtani** and **Abdullah Alsugair**

Civil Engineering Department, KSU-King Saud University, Riyadh 12372, Saudi Arabia;
kgahtani@ksu.edu.sa (K.A.-G.); amsugair@ksu.edu.sa (A.A.)
* Correspondence: abdul-kareem.e@hotmail.com

**Abstract:** Despite advancing Internet of Things (IoT) technologies, road projects often rely on inaccurate supplier data, making it difficult to determine the cost, quantity, quality, and transportation duration of the needed materials. The wrong choice of material suppliers can lead the supply chain to suffer losses, directly affecting the project's performance. In this regard, many studies have devised material logistics optimization models for road projects. However, the majority based their decisions on inaccurate or outdated data. This paper studies this gap by introducing a framework that utilizes IoT technologies and smart construction to feed optimization models with accurate and dynamically updated material data. This IoT-powered framework considers only quantitative criteria as input data to the integrated linear programming optimization model, precisely selected suppliers, and optimally calculated costs using *MS Excel Solver*. The results reveal that the framework is sensitive to any dynamic data updates and can achieve up to 40% material cost savings in real runtime. The paper demonstrates the proposed outline framework with a case study of planning an alternative road between Riyadh and Madinah cities in Saudi Arabia.

**Keywords:** construction management; automation; smart construction; IoT; optimization

## 1. Introduction

With the extensive support of the leadership, the Kingdom of Saudi Arabia (KSA) has become one of the leading countries in the communication and information technology sector, advancing to the eighth rank of the G-20 countries, according to the 2020 Saudi Press Agency (SPA) data [1]. The Chartered Institute for IT (BCS) [2] claimed that the ongoing fourth industrial revolution could be described as "a fusion of advances in artificial intelligence (AI), robotics, IoT" [3]. Locally, the KSA has an advanced IT position with almost 81% of businesses in KSA already using IoT platforms [3]. Nevertheless, the adoption is still limited in the construction sector, according to recent reports published by the Saudi Communications & Information Technology Commission [4]. The optimizing material logistics can utilize these advanced technologies to overcome the challenges and reduce road construction costs.

Road projects occupy enormous importance in the local economy, yet they usually present many inherent issues, especially material logistics costs [5]. As demonstrated in the next topic, various scholars have studied material logistics, including a few road projects, yet many models base their decisions on inaccurate or outdated data. One of the significant saving potentials inherent in road projects is optimizing material logistics using modern selection tools and dynamic data [6]. The incorrect selection can cause losses in the supply chain, hurting project performance directly. Thus, any enhancement of material supply in terms of time, quality, and cost would make a significant difference in project profitability, particularly long-distance road constructions.

Selecting a suitable material logistics model is essential for the rest work progress of construction projects and cost optimization. However, such models need readiness of

accurate and dynamically updated data through reliable sources to achieve meaningful material cost optimization. Thus, creating a dynamic data connectivity framework connected with an appropriate optimization method and tools, such as official open data (OD), global positioning system (GPS), blockchain (BC), mapping software (MSW), construction smart machines (CSM), and smart logistics trucks (SLT), is considered an intelligent step to support any logistics model. Such frameworks can provide project parties and stakeholders with accurate and reliable material data, such as duration, location, capacity, price, and type [7].

This paper proposes a smart framework for cost optimization that utilizes data connectivity (DC) technologies to feed the process with accurate, reliable, and dynamic material data to optimize material costs associated with road construction projects. The framework contains several smart elements. It can help decision makers reach rapid optimized decisions in terms of material cost with high dynamicity and accuracy input data, which is different from previous studies. Besides the developed linear programming (LP) model, enhancing the whole optimization process with precise and dynamic material data is the main contribution of this paper. Additionally, a case study of a planned project is used to demonstrate the proposed dynamic framework. The results reveal that the framework is sensitive to any dynamic data updates and can achieve up to 40% material cost savings in real runtime.

## 2. Literature Review

This section will discuss verities of studies related to the smart elements and technologies employed in the proposed framework. These elements are essential for accurate data preparation, cost optimization, and supplier selection process.

### 2.1. IoT Studies

Nowadays, the internet maintains a sense of logistics meaning. It enables us to search online data about the material, places, and people, utilizing search engines. In light of the internet, logistics can be defined as providing the right product in the right condition, amount, place, time, and cost [8]. The interaction between the internet and logistics in terms of IoT can deliver fruitful benefits. Logistics governance is broadly possible with the help of IoT.

The IoT is described as a DC network of physical smart objects or instruments that are autonomously capable of sensing, monitoring, managing, tracking, interacting with their surroundings, and collecting and exchanging data [4,9]. This intelligent DC technique enables the users to comprehend the various parameters and their effects, making decisions that benefit the system's operation [10]. According to Witkowski [11], up-to-date IoT data is a crucial solution for the logistics and transportation sectors since it can provide operational data on the location and status of items while also shortening the logistical process cycle and saving expenses. It is noteworthy that people fix their attention firmly upon communication when they talk about the IoT and neglect IoT protocols from their consideration. However, communication would fall without the interaction medium between IoT elements and devices. In this regard, IoT elements and devices use generally one of two IoT protocols. The first type is known as IoT data protocols which are used to connect low-power IoT devices. This type of protocol provides communication with hardware without the internet. The other protocols are recognized as IoT network protocols, which are typically used to link IoT elements and devices over the internet [12]. As a result, IoT may play a critical function in facilitating a dynamic data interchange across logistics and construction project participants. With this in mind, this function can provide the most up-to-date data for the selection and optimization of models, allowing them to identify the best suppliers and compute the optimal material costs.

## 2.2. Supplier Selection Criteria Studies

According to Cengiz [13], a proper supplier selection process seeks the ideal supplier who can serve the client with the right quality products or services at the right price, in the right proportions, and at the right time. Taherdoost and Brard [14] discovered that many companies still rely solely on pricing performance when deciding on a supplier. Nonetheless, several companies have adopted a more detailed, multi-criteria approach. Price, quality, and delivery were the most commonly used supplier evaluation factors [14,15]. Several scholars have classified supplier selection as a multiple criteria decision-making (MCDM) problem. However, the supplier selection problem may be solved using a simple linear programming (LP) approach when the selection criteria are limited and straightforward. Mukherjee [16] conducted a literature review and compiled the following list of the most commonly utilized supplier selection criteria.

- Cost
- Quality
- Delivery
- Service
- Supplier's profile
- Reliability
- Environment
- Responsiveness

- Logistical performance
- Commercial plans and structure
- Production
- Facility and technology
- Professionalism of salesperson
- Quality of relationship with vendor
- Risk factor
- Technology and capability

- Performance history
- R & D
- Mutual trust
- Easy communication
- Collaboration
- Annual demand
- Availability
- Supplier's willingness

## 2.3. Supplier Selection Methods

Selecting a suitable criteria model depends on the purchasing situation. Each client should decide on the criteria that correspond to their expectations and then create a ranking method to determine the best supplier [14]. However, there is no standard method for supplier selection. Because every supplier selection process is unique, companies employ various methods depending on their criteria, product, industry, and expectations. Tahriri et al. [17] discovered that the methods used are essential to the entire process and substantially affect the selection results. It is critical to comprehend why a company prefers one approach (or a mix of methods) over another. As a result, companies use the one that best matches their sector and expectations. Taherdoost and Brard [14] collected the most cited supplier selection methods in the literature and organized them into six categories, as follows:

1. Statistical/probabilistic (cluster analysis), such as fuzzy set theory.
2. Multi-attribute decision making (categorial methods), such as AHP, ANP, TOPSIS, MAUT, and outranking methods such as ELECTRE and PROMOTHEE.
3. Methods based on costs, such as ABC and TCO.
4. Mathematical programming (data envelopment analysis), such as LP, MOLP, and goal programming.
5. AI, such as CBR and ANN.

6.　Combined approaches, such as (MP + TCO, AHP + LP, MAUT + LP, ANP + TOPSIS, and fuzzy TOPSIS).

In this regard, Ali et al. [18] developed the LR-type fuzzy multi-objective model to solve realistic supplier selection problems and serve as a helpful tool for decision making in logistics management. Nazari-Shirkouhi et al. [19] developed a two-phase fuzzy multi-objective LP model for supplier selection with multiple pricing levels and products. Liao and Kao [20] performed a two-stage model for selecting the optimized suppliers in a watch manufacturing company by utilizing a fuzzy (TOPSIS) method with fuzzy triangular numbers and multi-choice goal programming (MCGP). Jadidi et al. [21] solved a multi-objective supplier selection problem, using a normalized goal programming model with specified goals and weights. Uygun et al. [22] integrated an approach combining ANP and decision-making trial and evaluation laboratory (DEMATEL) methods to evaluate the outsourcing providers in the telecommunication industry. The DEMATEL technique determined the criteria interdependences and the fuzzy ANP method for calculating the weights of the criteria and sub-criteria. Astanti et al. [23] conducted a comparative analysis between AHP and fuzzy AHP in the supplier selection problem, using experts in the field who have more than 12 years of experience in this area. This study's fuzzy AHP approach was unnecessary, mainly whenever 'expert' respondents supported the decision-making process. In such a case, the AHP without the fuzzy technique is sufficient for making the decision.

### 2.4. Studies Related to LP Methods of Optimizing Material Supply Chain

Numerous research papers have studied the optimization of the material supply chain of road construction projects using different optimization techniques [24–26]. They discussed how to deliver material from suppliers (SPL) to destinations and reduce logistics costs using LP optimization models. Choudhari and Tindwani [6] developed an advanced LP model. They considered three phases of a raw material supply chain that allow additional processing and mixing at intermediate locations before final consumption. Nevertheless, because of a lack of sophisticated computer software/computing power, their model focused on representative data, emphasizing the method, and did not probably cover topics such as data preparation speed, accuracy, and data automation. They found that current optimization modeling techniques in road construction projects need to be enhanced with up-to-date data. Arayapan and Warunyuwong [27] developed standard optimization models for transport planning to improve the overall logistics cost, including intangible costs. However, the models could not measure inbound logistics regularly and systematically.

According to the literature review, the present trend of supplier selection is an integrated or hybrid approach of two or more selection methods. However, developing advanced and mixed methods, such as fuzzy TOPSIS, FAHP, ELECTRE, etc., were not the objective of the current study. The authors considered only quantitative criteria as input data to the recommended selection approach; thus, the LP method is adequate to calculate quantitative values. Fathi and Bevrani [28] noticed that LP models were recently employed widely to optimize resource allocation issues, such as material supply plans due to the modest grade of exponents used in the decision variables. However, LP models must be fed with up-to-date data to return accurate selection results. The next topics will cover the smart resources needed for a successful material selection framework.

### 2.5. Studies Related to Location Data Accuracy for Optimizing Material Supply Chain

Besides IoT and DC technologies, data preparation is considered a vital part of the optimization process in this research. Sammut et al. [29] determined that before data can be analyzed, they must be organized into an appropriate form.

Dynamic and accurate location data are considered vital values needed in the current paper. Many research papers have discussed the quick acquisition of mapping data to calculate the transportation time between material suppliers and road construction sites.

In this regard, Beskorovainyi et al. [30] designed mathematical models for optimizing transportation routes in closed-loop logistics systems, considering several topological and functional constraints. The model does not consider the data accuracy of material and transportation prices. Zhao et al. [31] devised a system for assessing the consistency of GPS-based logistics data on speed, elevation, and position from three separate receivers. Cedillo-Campos et al. [32] delivered an integrated method to reliable measure variability on travel times and its impact on transportation flexibility. Devlin et al. [33] evaluated the positional accuracy of the dynamic non-differential global positioning system (non-DGPS) for tracing trucks in Ireland. Somplak et al. [34] applied a data reconciliation method that utilizes recording movement vehicles to mine the data. The database gathers the start and endpoint of the way (GPS coordinates) and the entire duration. Based on the earlier studies, feeding optimization models with accurate mapping data can enhance material logistics costs by selecting the lowest overall cost alternatives. However, these studies concentrated on obtaining travel data (routes and times) and did not utilize "dynamic" data to optimize construction material costs or supplier selection.

### 2.6. Studies Related to Data Validation for Optimizing Material Supply Chain Using BC

Besides the accuracy of location data, data validation (such as price, cost, and quality) is crucial for successful LP optimization models. Recent technology, such as BC, is a promising tool that impacts material data validation. Several recent studies demonstrated the application of BC techniques to construction management. Some of these studies have covered this technique generally, whereas the others have concentrated on a particular area. For example, Farouk and Darwish [35] introduced a new framework that integrates supplier relationship management (SRM) with customer relationship management (CRM) in an application suite by BC technology. Of note, one of the applications of BC using smart contracts, which is a promising technology, changes the future of project managing. In this regard, Penzes [36] illustrated that smart contracts could make supply chain management more straightforward and transparent through the lens of BC. Hence, BC can play a significant role in the proposed framework. For example, to be sure that the bought items (e.g., subbase stones) are fair trade, you would need a record of all transactions and events of the life of that item. That would be possible with BC, which is different from the centralized and traditional methods [37]. Accordingly, BC techniques can ensure data validation for optimization models and help develop a reliable framework for material selection, including cost optimization.

### 2.7. Studies Related to CSM in Optimizing Material Supply Chain

According to Japan Gov statistics [38], CSM is one of the most recent uses of sensor technology for construction machines and is regarded as a promising construction approach. One of Japan's leading construction machinery companies, Komatsu, was the first to offer this remarkable idea. All of the company's construction machinery is wirelessly connected, allowing the company's KOMTRAX management system to monitor and handle the data collected by the equipment. The system collects information, such as location, operation status, and fuel level, using built-in sensors to efficiently manage operations, optimize fuel consumption, forecast machine issues, and more. Thus, the CSM is a vital element of the current smart framework.

### 2.8. Studies Related to ICT Technology CSM in Optimizing Material Supply Chain

Komatsu introduced new ICT technology into its construction machinery in 2008 with the launch of its Autonomous Haulage System (AHS), the first automated operating system for super-large dump trucks in the world. The firm then developed its smart construction Platform in 2015, which utilizes ICT to dynamically integrate not only construction machines, but all phases of the construction process to achieve total optimization. Smart construction is already in operation at over 3300 locations across Japan [38].

*2.9. Smart Networks and IoT-Based Real-Time Production Logistics*

The relationship between sustainable urban governance networks and Internet of Things-based real-time production logistics as regards construction road projects has been covered in several recent studies. Evans and Horak [39] performed assessments about how data-driven IoT systems and machine learning-based analytics can interact and exchange big data by utilizing networked and integrated sustainable urban technologies, such as sensing infrastructures and smart city solutions. They anticipated a growing future of utilizing big data sources to link communities and cities and to improve local government technological knowledge. Nica et al. [40] presented an exploratory review of the current research on IoT-based real-time production logistics and found that by employing various sensors, the logistical status of front-line machines and the operating setting data can be controlled sufficiently. Popescu et al. [41] came up with analyses concerning smart IoT systems and conducted that operational reliability can be improved by deploying a machine condition monitoring system. They found that fast data supplied by IoT can construct manufacturing environments that can optimize outputs into digitalized and networked systems across a smart framework that assists in decision making by use of massive real-time data and interaction and teamwork with equipment, sensors, and operators.

*2.10. Discussion of the Gap of Knowledge in This Study*

Briefly, the lack of data accuracy and dynamicity is the gap of knowledge of the selection and optimization models demonstrated in the above-given studies. Their main objective was to minimize the total cost, including ordering, inventory holding, purchasing, and transportation costs. Consequently, the main contribution of this paper is assembling a smart framework with fourth industry revolution techniques to ensure data accuracy, dynamicity, and validation to achieve accurate outputs of the employed LP optimization model.

Next, a conception of the smart framework is demonstrated by a case study to aid optimization and selection models with accurate, updated, validated, and dynamic supplier data. Besides the selection model, this framework consists of multiple smart elements such as OD source, mapping software, BC technology, and more to maintain dynamic data validation for the material cost optimization and calculation process.

**3. Flowchart of the Proposed Smart Framework for Logistics Cost Optimization**

As presented in Figure 1, the framework consists of two main components, Sections A and B. Section A utilizes IoT technologies and network protocols to collect and validate all input data needed to calculate the discount cost ($C_{ji}$) initiated between the current workstation (WST) and the potential supplier (SPL). Section B receives ready calculated $C_{ji}$ and employs it to proceed with the MS Excel Solver and LP Equations (2)–(5) to compare and select the best SPL alternatives. More explanation about the two sections can be reached in the following paragraphs.

*3.1. Section A: Collecting and Validating of Dynamic Input Data of Material Logistics Cost*

This section consists of four main modules of input data. The first module delivers by CSM and represents input data of the current WST position: demand of material, GPS coordinates, and material specifications. The second module describes an OD, which contains all accurate and dynamically updated data about road construction materials and all officially registered SPLs. The third module contains two aided tools that help to define distance and validate supplier data. The first aided tool utilizes MSW to determine the distance between any two locations considered in this framework. The second aided tool uses BC to validate construction material data (purchase and transportation prices). The fourth module (central module) considers all data acquired over the IoT network protocols from modules (1 to 3) to calculate the discount cost ($C_{ji}$) that arises between current WST and potential SPLs to purchase and transport construction materials. This module utilizes DC technologies, such as IoT, to collect dynamic data from other modules. The output data

of the central module flow as input data to the MS Excel Solver (LP optimization algorithm) in Section B.

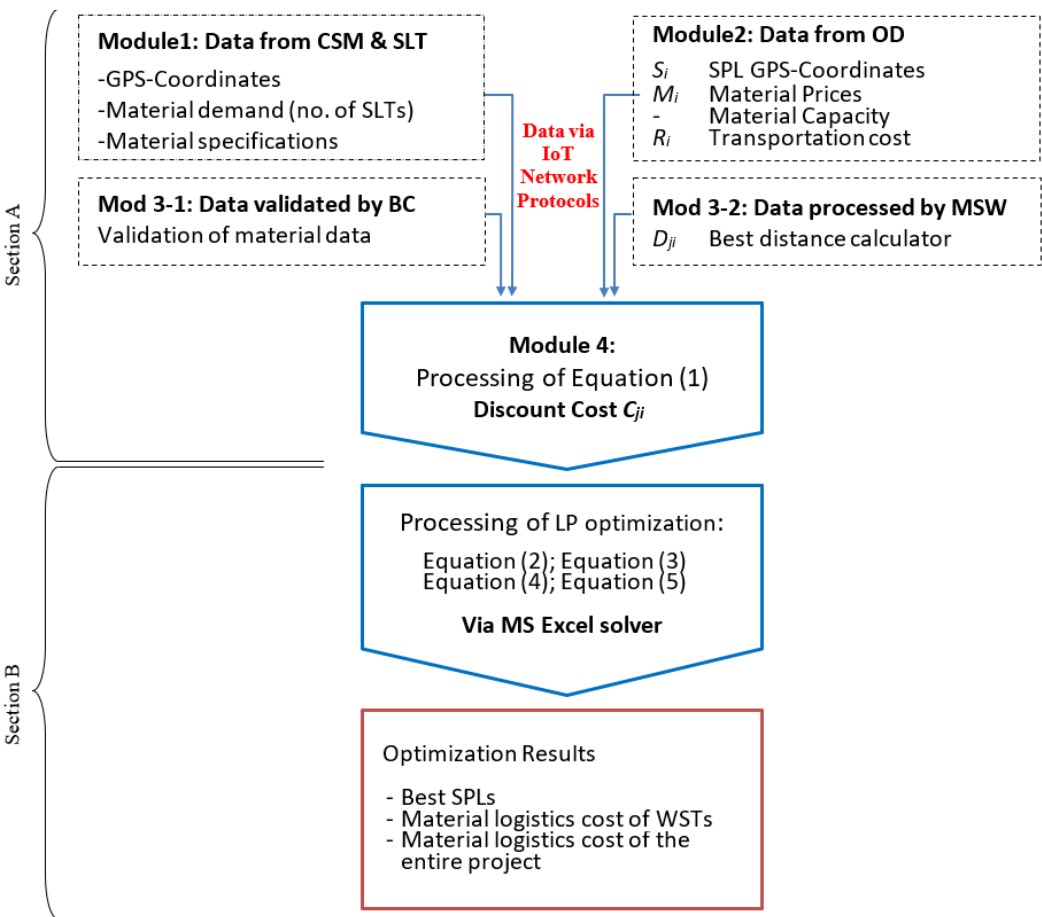

**Figure 1.** The flowchart of the proposed smart framework.

In detail, the first module provides the central module with the requirements of the current WST. These requirements (or input data) are fed automatically by CSM (client's machine) that works at the current GPS coordinates of WST. The input data in this part cover WST requirements, such as the following:

- Material demand volume (denoted by variable $T_j$) is required for the current WST. The active CSM working at the current position predicts this material demand volume. The variable counts the number of fully-loaded SLTs needed to cover the demand of this site.
- GPS coordinates of the current WST (latitude and longitude). These data are fed permanently by the active CSM and used by the MSW to determine the distance between the current WST and potential SPLs.
- Type of construction materials required to the current WST.

The second module is termed OD and provides reliable data about available material SPLs. It covers such data as follows:

- GPS coordinates of trusted SPL (exact latitude and longitude);
- Material transportation price per km ($R_i$);
- Material types available by SPLs (M-types, such as M1, M2, and M3);
- Capacity: the maximum count of fully loaded SLTs that SPL can send (the quantity unit of the transported material is measured by the count of fully loaded SLTs, not m$^3$);
- The material price, which is updated dynamically by SPLs ($M_i$).

The third module consists of two aided tools, namely BC and MSW. CSM can initialize the MSW to calculate the expected distance ($D_{ji}$) between current WST ($P_j$) and potential material SPLs ($S_i$). The MSW employs the GPS coordinates of the current WST ($P_j$) and potential SPLs ($S_i$) to calculate the accurate logistics distance ($D_{ji}$). This calculation process is performed repeatedly by CSM every kilometer along the WST. The purpose of repeating the $D_{ji}$ calculation is to consider all the suppliers' options and find the fastest and shortest $D_{ji}$. The logistics distance ($D_{ji}$) is a genuine variable of the discount cost ($C_{ji}$) calculated in the 4th module. On the other hand, BC is responsible for validating material data ($R_i$ and $M_i$) imported from the OD.

The fourth module in section A (central module) utilizes Equation (1) to advance discount cost ($C_{ji}$) based on data collected from other modules. All variables mentioned in section A (GPS coordinates, $R_i$, M-types, capacity, and $M_i$) flow in the following central Equation (1) to determine the discount cost ($C_{ji}$) of material procurement and transportation between current WST ($P_j$) and potential material SPLs ($S_i$).

$$C_{ji} = T_j \cdot D_{ji} \cdot R_i + T_j \cdot V_T \cdot M_i \tag{1}$$

where

$T$:　Material demand (number of fully loaded SLTs needed for current WST);
$D$:　Accurate logistics distance (derived from MSW);
$R$:　Updated transportation cost per km (imported from OD and proved by BC);
$V$:　Constant SLT volume equals 30 m$^3$ (max. 24 ton);
$M$:　Material procurement cost per m$^3$ (imported from OD and proved by BC).

One essential objective of this study is realized by linking all section A data together through IoT and DC techniques.

### 3.2. Section B: Applying LP Optimization Functions by MS Excel Solver

This section performs the following LP steps to select appropriate SPLs and optimize the material logistics cost for the entire project.

- Conceptualize sourcing of raw material for road construction as a logistics optimization problem.
- Formulate the LP model for the given problem statement and implied resource constraints.
- Solve the LP formulation by an optimization solver.
- Interpret the final output of the solver to determine the quantity to be procured from each supplier and distributed through various demand locations of the road.

For better understanding, a simplified sketch figure of the proposed LP model was created to develop the idea of calculation variables. Figure 2 depicts the pictorial representation of the problem. Each WST ($P_j$) can cover its demand from one or more SPLs ($S_i$) and vice versa. At the end phase, the model creates numerous relationships (decision variable $X_{ji}$) among WSTs and SPLs.

The above-illustrated sketch shows two perspectives of relationships ($X_{ji}$) arising between system nodes (SPLs and WSTs). The first represents material shipping from a single SPL to one or more WSTs. The sketch shows that SPL $S_9$ is shipping materials to all WSTs ($P_1$ to $P_{10}$). The second perspective represents the situation when the material is shipped from multiple SPLs to a single WST, such as SPLs $S_{1-20}$ to WST $P_1$, as depicted in Figure 2. Moreover, another type of relationship is the unit discount cost ($C_{ji}$) arising between system nodes, such as $C_{1,1}$, forming the unit discount cost between $P_1$ and $S_1$.

Under those circumstances, the proposed optimization model can be formulated simply by using LP techniques. The related formulations and corresponding variables of the model are defined in Table 1.

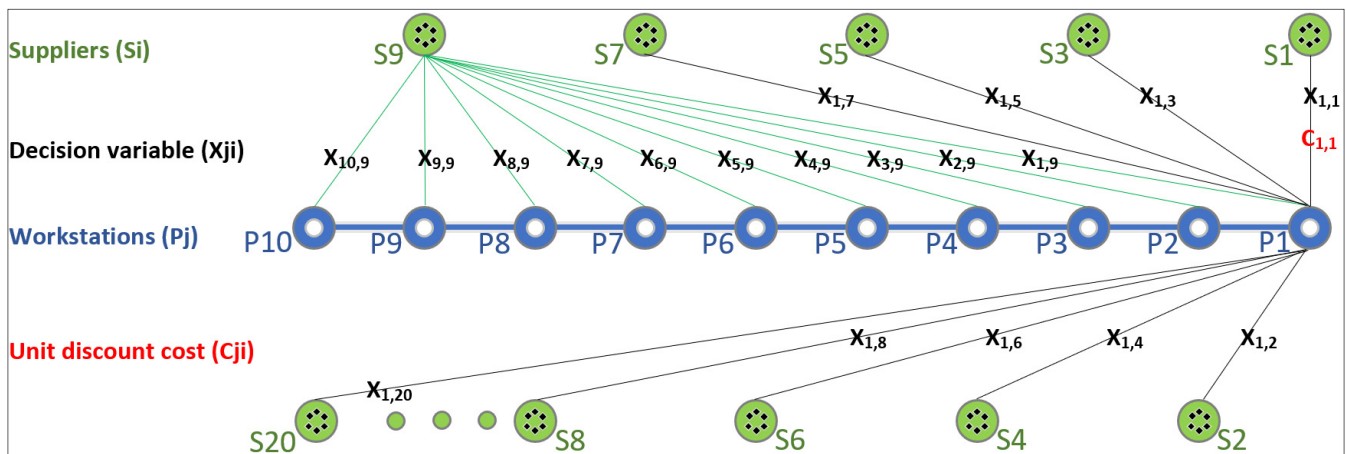

**Figure 2.** Sketch of the proposed material logistics LP model.

**Table 1.** Description of notations used in LP formulation.

| Notation | Description |
|---|---|
| **Index** | |
| i | SPL index (from 1 to $I$) |
| j | WST index (from 1 to $J$) |
| **Decision variables** | |
| $x_{ij}$ | Unit quantity of raw material shipped from $S_i$ to $P_j$ |
| **Parameters** | |
| $I$ | Total number of SPLs |
| $J$ | Total number of WSTs |
| $c_{ij}$ | Discount cost of transporting single unit of material from $S_i$ to $P_j$ |
| $\rho_i$ | The maximum supply capacity of raw material at source $S_i$ (per SLT) * |
| $\mu_j$ | The total demand of raw material at WST $P_j$ (per SLT) * |

* The quantity unit of transported material counted by number of full loaded SLTs, not m$^3$.

Accordingly, the mathematical formulations can be expressed as follows based on the LP conditions [42].

Objective function (Min Cost)

$$\sum\sum C_{ij} \cdot X_{ij} \tag{2}$$

Subject to:

for i = 1, ... I

$$\sum X_{ij} \leq \rho_i \tag{3}$$

for j = 1, ... J

$$\sum x_{ij} \geq \mu_j \tag{4}$$

$$x_{ij} \geq 0 \tag{5}$$

All symbols demonstrated above in the LP equations are termed in Table 1. These equations are linear with the decision variables. Thus, LP is an appropriate method for this problem. Equation (2) states the objective function, which minimizes the total supply cost of shipping desired quantity of material to the project. This cost contains the unit price and transportation cost as indicated previously. Equation (3) is the first constraint expression which imposes that the demanded material volume cannot exceed the SPL capacity of material ($\rho_i$). In contrast, the second constraint, Equation (4), imposes that the shipped volume of material must meet the demand of each WST ($\mu_j$). The last constraint, Equation (5), satisfies the non-negativity conditions and assures that all associated decision variables can be zero or take positive values.

## 4. Case Study: Enhancing Material Logistics Cost with a Smart Framework

### 4.1. Case Study Description

This case study considered a planned road construction project between two major cities in Saudi Arabia, namely Riyadh and Madinah, as presented in Figure 3. These two cities are vital in Saudi Arabia. Riyadh is the country's capital, having a population of more than 7 million people. In addition, Riyadh is a hub business for major local and international companies. On the other side, Madinah city is a holy city that attracts millions of Muslim visitors annually. The demand for connecting these two cities has increased yearly, and there is a need to increase the access road between them.

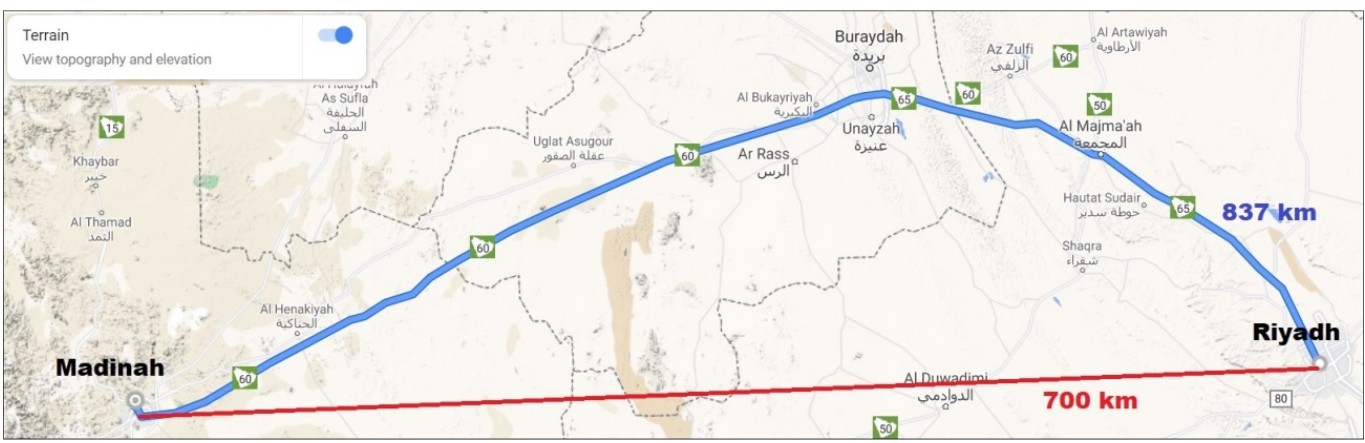

**Figure 3.** The newly planned road (in red) and the current road (in blue) (Google Maps | Terrain).

The newly planned road (indicated by the red line in Figure 3) is around 137 km shorter than the existing one (blue line), which benefits this vital project. The Google function called "Terrain" considers the topographical situation of the new road, as shown in Figure 3. This function confirms there are no significant topographical barriers (such as elevation, mountains, lakes, rivers, etc.) that may affect the logistics of the road work sites.

Moreover, the planned project intersects with many local routes that offer access paths for transportation trucks that supply the project with the required materials. Figure 4 shows some local routes (in red) intersecting the project line (in black) according to the data gained from Saudi GASGI [43]. In this case study, MSW utilized these access routes to measure the distance between suppliers (SPL $S_i$) and workstations (WST $P_j$).

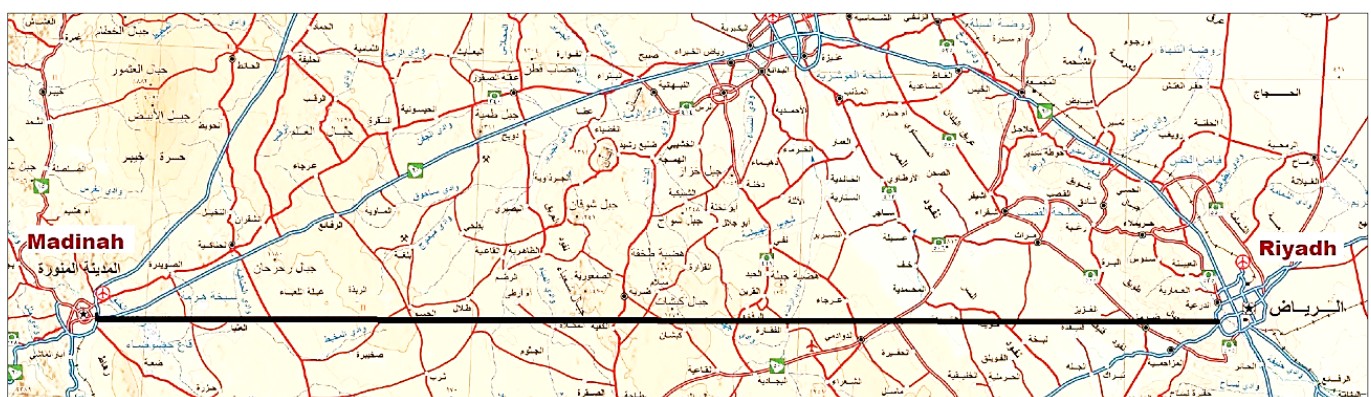

**Figure 4.** Local routes (in red).

The planned project scope includes building a three-lane international highway road for a 700 km stretch with 10 WSTs. Raja [44] explained the standard structure of road layers, in which the construction layer comes directly under the road surface. In the current case,

each road direction is almost 11 m wide. In addition, it reaches an average depth of 50 cm under the surface courses along with the project.

However, to demonstrate the cost-saving of material supply in this study, only one type of material was adopted, namely, crushed stone. Crushed stone is a widely accessible substance in Saudi Arabia and forms an essential stone layer for the highest traffic volume category in the country. Filling this layer with stones delivered by carefully selected SPLs presents a real challenge that faces project decision makers repeatedly. For this reason, road projects often rely on inaccurate SPL data, which makes it difficult to determine the cost, quantity, quality, and transportation duration of the demanded materials.

*4.2. Applying Case Study with the Proposed Framework*

The authors proposed a comprehensive framework that assures a smart and fast DC medium to link trusted material SPLs with their clients to overcome the challenges mentioned above. With this in mind, the smart framework imports SPL data validated regularly by the Ministry of Industry and Mineral Resources (MIM). The framework utilizes many smart elements such as OD, LP, BC, MSW, CSM, SLT, and sensors that interact under IoT techniques. This framework can offer accessible and reliable supplier data in the end phase in case of direct adoption from MIM.

The following example illustrates the connectivity mechanism that interacts among all the aforementioned smart elements. Each active CSM uses built-in sensors and an optimization algorithm to predict the material demand for the current project stage. Then, the machine uses the IoT network protocols to connect automatically with OD to reach validated supplier data. The CSM receives over the IoT network protocols all validated material data and calculates the discount cost $C_{ji}$ of each SPL. Based on quantitative selection criteria stored in the machine's software, the CSM processes LP optimization (MS Excel Solver) and selects the best SPL among all available material SPLs. On the other hand, the SPL's system receives a request from the CSM that orders a specific material type/amount. Then the SPL sends several SLTs to satisfy the CSM demand. As the roadwork progresses, the WST-coordinates also move. Consequently, the distance between the material SPLs and the WST changes simultaneously. With this in mind, the optimization of material logistics needs to be readjusted frequently by CSM. This step recalculates the new material volume and the best SPLs based on the newly recognized WST coordinates. Accordingly, the optimization algorithm stored in the CSM is responsible for calculating a new optimized material cost. All fetched data are validated simultaneously by smart elements facilitated by IoT technologies.

In this paper, the authors have intentionally concentrated on one single factor to demonstrate the reliability of the proposed framework. This includes only factors that affect the discount cost $C_{ij}$ of material logistics (such as material cost and transportation cost). Accordingly, the requested material in this study is limestone size = 0.075/37.5 mm. Moreover, SLT size = 30 m$^3$ with a maximum 24 Ton weight are considered to haul the needed material from the trusted local quarries. Such information can be conducted through the OD connected by IoT.

The framework uses MSW, such as Google maps, which obtains accurate GPS coordinates from the OD and returns distance value as input data to the LP model. Through this process, CSM can obtain the best transportation cost and duration of each SLT hauling material to the project. Table 2 illustrates an example of the data structure offered by the OD. Due to privacy rights, all suppliers' names are omitted, as shown in the table.

**Table 2.** An example of dynamic data offered by OD.

| Field Name | SPL $S_1$ | SPL $S_2$ | SPL $S_3$ | ... $S_i$ |
|---|---|---|---|---|
| location (city) | Jubail | Riyadh | Jeddah | ... |
| GPS (lat., long.) | 27.15, 49.2 | 24.98, 46.99 | 21.47, 39.39 | ... |
| raw material type | M1 | M3 | M1 | ... |
| material description | limestone | silica sand | limestone | ... |
| capacity (per SLT) | 210,372 | 135,621 | 77,235 | ... |
| price (SAR/SLT) | SAR 830.00 | SAR 720.00 | SAR 980.00 | ... |
| available transporters (SLTs) | 50 | 33 | 75 | ... |
| transportation price (per SLT/km) | SAR 3.50 | SAR 4.00 | SAR 4.30 | ... |
| quotation update time | 16/9/21 7:16 | 19/9/21 13:52 | 18/9/21 0:43 | ... |

Figure 5 illustrates a microscale location map of the road WSTs considered in this project. The WSTs are denoted by variable $P_j$, which varies the index j from WST to WST. In detail, WST ($P_1$) is the start point of the planned project and refers to WST of Riyadh City. All following WSTs ($P_2$–$P_{10}$) are secured with logistics routes (as depicted in Figure 4) and ended with Madinah WST ($P_{10}$). Each WST could have unlimited logistics alternatives ($X_{ji}$), as shown in Figure 5 and later called decision variables. Decision variables ($X_{ji}$) represent the relationship between WST ($P_j$) and SPL ($S_i$), where the index j and i refer to WST and SPL, respectively. Thus, CSM has a wide range of supply alternatives that can be selected among all recognized suppliers in the country. Note that the grey line between WSTs and SPLs (for instance, $S_{36}$ with $P_{10}$, etc.) forms the mathematical relationship or the decision variable ($X_{36,10}$), not the actual route distance.

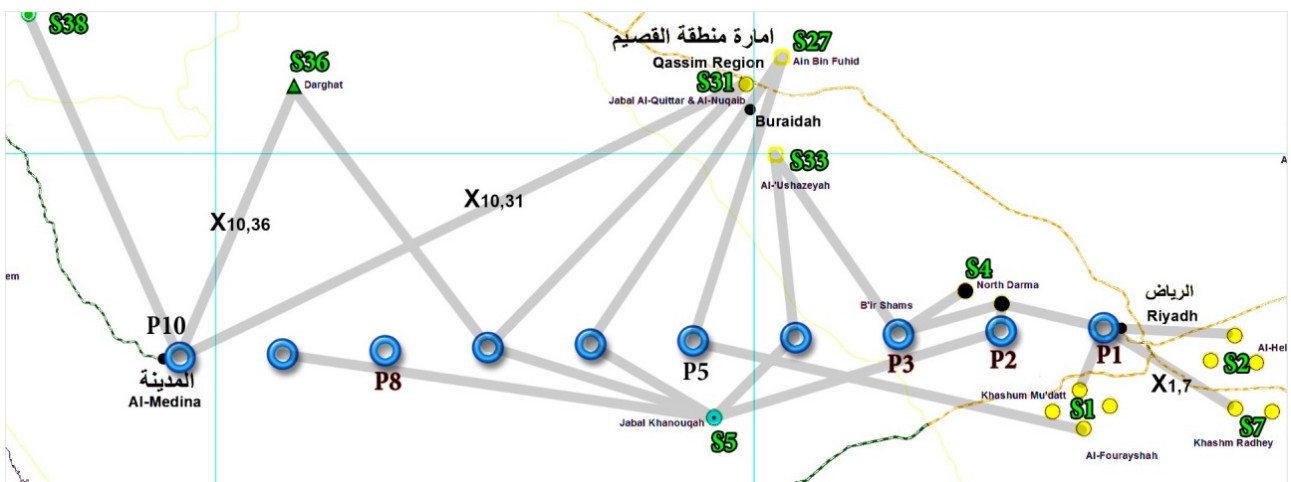

**Figure 5.** Project nodes WSTs ($P_j$), SPLs ($S_i$), and their decision variables ($X_{ji}$) [45].

Considering the decision variables ($X_{ji}$ lines colored grey) between material SPLs and project WSTs, a two-dimensional matrix with 20 (SPLs) and 10 (WSTs) was created to form a range of 200 decision variables ($X_{1,1}$; $X_{20,10}$) in this study. Despite the vast number of local SPLs in Saudi Arabia, the authors have intentionally decreased the considered SPLs ($S_i$) count to demonstrate the concept more easily. Generally, the details mentioned earlier and influence factors that impact road construction are considered in the following sections. These factors are essential to developing a framework that can reduce the costs associated with the project material logistics.

*4.3. Results and Discussion of the Case Study*

Considering the above-discussed mathematical formulations, an *MS Excel Solver* was fed dynamically with the data collected and validated via IoT in Section A. The data contain values (such as GPS coordinates, material prices, demand, capacities, etc.). Using this data, the CSM calculates Equation (1) and returns the discount cost $C_{ji}$ related to the demanded

material needed for the current WST. Next, the CSM sends the result ($C_{ji}$) to the solver that processes Equations (2)–(5) and calculates the optimal logistics plan.

The expressions, such as cost-minimizing objective function, constraints, and decision variable fields, are defined and set at appropriate cells in *MS Excel Solver*, as shown in Figure 6. An appropriate solving method, namely Simplex LP, was also nominated as the default method through the software. All decision variables were set as integers due to the nature of the transportation problem. The unconstrained decision variables were set as non-negative to satisfy the condition of non-negativity. Figure 6 illustrates the input and output data representation in an excel spreadsheet, which comprises three main sub-tables needed for the *Solver* calculations.

**Cij: Discount cost between suppliers and workstations**

| Si | P1 | P2 | P3 | P4 | P5 | P6 | P7 | P8 | P9 | P10 |
|---|---|---|---|---|---|---|---|---|---|---|
| S1 | 2281 | 2069 | 2089 | 1853 | 1769 | 1917 | 2285 | 2537 | 2585 | 2765 |
| S2 | 1050 | 1272 | 1609 | 1718 | 2024 | 2318 | 3041 | 3097 | 3292 | 3440 |
| S3 | 908 | 1088 | 1407 | 1658 | 1909 | 2152 | 2670 | 2838 | 2878 | 2997 |
| S4 | 720 | 1021 | 1553 | 1863 | 2448 | 2682 | 3609 | 3780 | 3951 | 4154 |
| S5 | 742 | 890 | 1150 | 1309 | 1513 | 1711 | 2156 | 2239 | 2321 | 2420 |
| S6 | 769 | 930 | 1213 | 1388 | 1612 | 1825 | 2320 | 2401 | 2500 | 2608 |
| S7 | 888 | 1098 | 1426 | 1713 | 2003 | 2261 | 2818 | 2960 | 3064 | 3192 |
| S8 | 704 | 958 | 1408 | 1784 | 2164 | 2480 | 3259 | 3418 | 3547 | 3718 |
| S9 | 1196 | 1463 | 1943 | 2349 | 2734 | 3099 | 3964 | 4120 | 4271 | 4456 |
| S10 | 695 | 838 | 1098 | 1316 | 1536 | 1732 | 2209 | 2262 | 2346 | 2443 |
| S11 | 1056 | 1197 | 1459 | 1681 | 1879 | 2075 | 2556 | 2622 | 2721 | 2840 |
| S12 | 992 | 1146 | 1374 | 1562 | 1752 | 1922 | 2334 | 2379 | 2452 | 2535 |
| S13 | 919 | 1048 | 1282 | 1480 | 1702 | 1848 | 2268 | 2344 | 2420 | 2510 |
| S14 | 1149 | 1363 | 1749 | 2076 | 2386 | 2680 | 3376 | 3501 | 3627 | 3772 |
| S15 | 1065 | 1227 | 1665 | 2031 | 2375 | 2704 | 3537 | 3500 | 3840 | 4007 |
| S16 | 2443 | 2126 | 1607 | 1211 | 990 | 1273 | 2206 | 2184 | 2580 | 3064 |
| S17 | 1024 | 1112 | 1409 | 1654 | 1907 | 2112 | 2672 | 2647 | 2887 | 2999 |
| S18 | 1043 | 1229 | 1724 | 2140 | 2531 | 2905 | 3845 | 3808 | 4194 | 4383 |
| S19 | 1885 | 1743 | 1515 | 1337 | 1273 | 997 | 959 | 1007 | 1131 | 1387 |
| S20 | 1207 | 1239 | 1571 | 2107 | 2367 | 2727 | 3591 | 3551 | 3943 | 4255 |

| Si | P1 | P2 | P3 | P4 | P5 | P6 | P7 | P8 | P9 | P10 | Total Supply | Constraints (Trucks) |
|---|---|---|---|---|---|---|---|---|---|---|---|---|
| S1 | 1 | 1 | 1 | 1 | 1 | 1 | 1 | 1 | 1 | 1 | 10 | ≤ 66,000 |
| S2 | 1 | 1 | 1 | 1 | 1 | 1 | 1 | 1 | 1 | 1 | 10 | ≤ 46,200 |
| S3 | 1 | 1 | 1 | 1 | 1 | 1 | 1 | 1 | 1 | 1 | 10 | ≤ 75,900 |
| S4 | 1 | 1 | 1 | 1 | 1 | 1 | 1 | 1 | 1 | 1 | 10 | ≤ 23,100 |
| S5 | 1 | 1 | 1 | 1 | 1 | 1 | 1 | 1 | 1 | 1 | 10 | ≤ 132,000 |
| S6 | 1 | 1 | 1 | 1 | 1 | 1 | 1 | 1 | 1 | 1 | 10 | ≤ 59,400 |
| S7 | 1 | 1 | 1 | 1 | 1 | 1 | 1 | 1 | 1 | 1 | 10 | ≤ 62,700 |
| S8 | 1 | 1 | 1 | 1 | 1 | 1 | 1 | 1 | 1 | 1 | 10 | ≤ 50,820 |
| S9 | 1 | 1 | 1 | 1 | 1 | 1 | 1 | 1 | 1 | 1 | 10 | ≤ 102,300 |
| S10 | 1 | 1 | 1 | 1 | 1 | 1 | 1 | 1 | 1 | 1 | 10 | ≤ 112,200 |
| S11 | 1 | 1 | 1 | 1 | 1 | 1 | 1 | 1 | 1 | 1 | 10 | ≤ 198,000 |
| S12 | 1 | 1 | 1 | 1 | 1 | 1 | 1 | 1 | 1 | 1 | 10 | ≤ 118,800 |
| S13 | 1 | 1 | 1 | 1 | 1 | 1 | 1 | 1 | 1 | 1 | 10 | ≤ 95,700 |
| S14 | 1 | 1 | 1 | 1 | 1 | 1 | 1 | 1 | 1 | 1 | 10 | ≤ 138,600 |
| S15 | 1 | 1 | 1 | 1 | 1 | 1 | 1 | 1 | 1 | 1 | 10 | ≤ 66,000 |
| S16 | 1 | 1 | 1 | 1 | 1 | 1 | 1 | 1 | 1 | 1 | 10 | ≤ 89,100 |
| S17 | 1 | 1 | 1 | 1 | 1 | 1 | 1 | 1 | 1 | 1 | 10 | ≤ 105,600 |
| S18 | 1 | 1 | 1 | 1 | 1 | 1 | 1 | 1 | 1 | 1 | 10 | ≤ 128,700 |
| S19 | 1 | 1 | 1 | 1 | 1 | 1 | 1 | 1 | 1 | 1 | 10 | ≤ 52,800 |
| S20 | 1 | 1 | 1 | 1 | 1 | 1 | 1 | 1 | 1 | 1 | 10 | ≤ 33,000 |
| **Total Demand** | 20 | 20 | 20 | 20 | 20 | 20 | 20 | 20 | 20 | 20 | 200 | |
| | = | = | = | = | = | = | = | = | = | = | | **Objective Fuction** |
| Demand | 25,667 | 10,267 | 12,833 | 20,533 | 25,667 | 15,400 | 17,967 | 10,267 | 7700 | 12,833 | Total Cost | 431,924 |
| | | | | | | | | | | | | Goal: Min |

**Xij: Material transport from suppliers to workstations**

| Si | P1 | P2 | P3 | P4 | P5 | P6 | P7 | P8 | P9 | P10 |
|---|---|---|---|---|---|---|---|---|---|---|
| S1 | X11 | X12 | X13 | X14 | X15 | X16 | X17 | X18 | X19 | X110 |
| S2 | X21 | X22 | X23 | X24 | X25 | X26 | X27 | X28 | X29 | X210 |
| S3 | X31 | X32 | X33 | X34 | X35 | X36 | X37 | X38 | X39 | X310 |
| S4 | X41 | X42 | X43 | X44 | X45 | X46 | X47 | X48 | X49 | X410 |
| S5 | X51 | X52 | X53 | X54 | X55 | X56 | X57 | X58 | X59 | X510 |
| S6 | X61 | X62 | X63 | X64 | X65 | X66 | X67 | X68 | X69 | X610 |
| S7 | X71 | X72 | X73 | X74 | X75 | X76 | X77 | X78 | X79 | X710 |
| S8 | X81 | X82 | X83 | X84 | X85 | X86 | X87 | X88 | X89 | X810 |
| S9 | X91 | X92 | X93 | X94 | X95 | X96 | X97 | X98 | X99 | X910 |
| S10 | X101 | X102 | X103 | X104 | X105 | X106 | X107 | X108 | X109 | X1010 |
| S11 | X111 | X112 | X113 | X114 | X115 | X116 | X117 | X118 | X119 | X1110 |
| S12 | X121 | X122 | X123 | X124 | X125 | X126 | X127 | X128 | X129 | X1210 |
| S13 | X131 | X132 | X133 | X134 | X135 | X136 | X137 | X138 | X139 | X1310 |
| S14 | X141 | X142 | X143 | X144 | X145 | X146 | X147 | X148 | X149 | X1410 |
| S15 | X151 | X152 | X153 | X154 | X155 | X156 | X157 | X158 | X159 | X1510 |
| S16 | X161 | X162 | X163 | X164 | X165 | X166 | X167 | X168 | X169 | X1610 |
| S17 | X171 | X172 | X173 | X174 | X175 | X176 | X177 | X178 | X179 | X1710 |
| S18 | X181 | X182 | X183 | X184 | X185 | X186 | X187 | X188 | X189 | X1810 |
| S19 | X191 | X192 | X193 | X194 | X195 | X196 | X197 | X198 | X199 | X1910 |
| S20 | X201 | X202 | X203 | X204 | X205 | X206 | X207 | X208 | X209 | X2010 |

**Figure 6.** Material logistics data fed automatically to *MS Excel Solver*.

In detail, the first sub-table includes a pre-calculated unit discount cost ($C_{ji}$) of shipping one loaded SLT from SPL ($S_i$) to WST ($P_j$). The calculation to obtain the discount cost ($C_{ji}$) that arises between particular nodes, namely SPL ($S_{i=2}$) and WST ($P_{j=3}$), can be taken from Equation (1) as follows.

This is a numerical example of Equation (1):

$$C_{3,2} = T_{j=3} \cdot D_{3,2} \cdot R_{i=2} + T_{j=3} \cdot V_T \cdot M_{i=2}$$

where the GPS coordinates of nodes P3 and S2 are from other modules (1 and 2). Then, we calculate the exact distance between them, and

$j = 3$, which refers to WST $P_3$

$I = 2$, which refers to SPL $S_2$

$T_3 = 1$, which represents one full loaded SLT of limestone collected from WST $P_3$

$D_{3,2} = 242$ km, where this value is derived from MSW (Module 3), which imports

$R_2 =$ SAR 3.30 for one fully loaded SLT per km. These data were imported from Module (2) and validated by Module (3).

$V_T$ = 30 m³ SLT volume (max. 24 ton). This value is constant.

$M_2$ = SAR 27 per m³. Module (2) and (3) imported and validated these data.

By substituting all variables in equation (1) with the above collected values, a discount cost ($C_{3,2}$) of shipping one fully loaded SLT from $S_2$ to $P_3$ reached the following result:

$$C_{3,2} = 1 \text{ SLT} \cdot 242 \text{ km} \cdot \text{SAR } 3.30 + 1 \text{ SLT} \cdot 30 \text{ m}^3 \cdot \text{SAR } 27$$

$$C_{3,2} = \text{SAR } 1609 \text{ per one fully loaded SLT (delivered)}$$

The previous calculation procedure of unit discount cost is repeated among all other system nodes ($S_i$ and $P_j$) in this project. As previously stated, with varying the GPS coordinates of construction location, the active CSM will perform this calculation repeatedly to find the best SPLs and competitive material logistics costs. Repeating this calculation as a response for the variable GPS coordinates is more accurate. However, in this paper, we fixed the GPS coordinates of WSTs ($P_1$–$P_{10}$) to simplify the associated LP calculations. The results of all discount cost calculations of the fixed WSTs are given in the 1st sub-table of Figure 6. The discount cost $C_{3,2}$ can be seen in excel cell number D4 with the same result SAR 1609 as previously calculated to prove the given values.

Furthermore, the second sub-table nominates all possible decision variables ($X_{ji}$) needed to ship the one-unit quantity of construction material from any $S_i$ to any $P_j$. The 3rd sub-table contains *Solver* functions and utilizes all data imported from other sub-tables. The *Solver* proceeds all LP algorithms and returns optimized results.

Since the road has 700 km, it is required to have thousands of WSTs and GPS coordinates in response to the continuous movement of the active CSM. This arrangement requires many calculations requiring powerful software to cover all the possibilities raised between different project nodes. Hence, the concept of 10 WSTs was used in this case study with a fixed distance (70 km) between them. These 10 WSTs will be connected with 20 SPLs and form a matrix of 200 values (cost data) that may arise between those nodes (i.e., 10 WSTs · 20 SPLs = 200 cost data). The second sub-table in Figure 6 presents these values in tabulated form.

The main *Solver*'s sub-table satisfies all mathematical formulations of LP Equations (2)–(5). The material demand limitations for WSTs (from $P_1$ to $P_{10}$) were inserted in demand cells (Cells, N25:W25). Similarly, the material capacity constraints of 20 SPLs were added to column Z (Cells, Z3:Z22). Respectively, demand and capacity values were fed to the *Solver* by CSM and OD. Both data values were already prepared in Table 3 to be used in the LP optimization process of the *Solver*.

**Table 3.** WST demands and SPL capacities (measured in SLTs).

| WST $P_j$ | Demand (SLTs) | SPL $S_i$ | Capacity (SLTs) | SPL $S_i$ | Capacity (SLTs) |
|---|---|---|---|---|---|
| $P_1$ (depth = 0.5 m) | 25,667 | $S_1$ | 66,000 | $S_{11}$ | 198,000 |
| $P_2$ (0.2 m) | 10,267 | $S_2$ | 46,200 | $S_{12}$ | 118,800 |
| $P_3$ (0.25 m) | 12,833 | $S_3$ | 75,900 | $S_{13}$ | 95,700 |
| $P_4$ (0.4 m) | 20,533 | $S_4$ | 23,100 | $S_{14}$ | 138,600 |
| $P_5$ (0.5 m) | 25,667 | $S_5$ | 132,000 | $S_{15}$ | 66,000 |
| $P_6$ (0.3 m) | 15,400 | $S_6$ | 59,400 | $S_{16}$ | 89,100 |
| $P_7$ (0.35 m) | 17,967 | $S_7$ | 62,700 | $S_{17}$ | 105,600 |
| $P_8$ (0.2 m) | 10,267 | $S_8$ | 50,820 | $S_{18}$ | 128,700 |
| $P_9$ (0.15 m) | 7700 | $S_9$ | 102,300 | $S_{19}$ | 52,800 |
| $P_{10}$ (0.25 m) | 12,833 | $S_{10}$ | 112,200 | $S_{20}$ | 33,000 |

Before starting the *Solver*, all decision variables ($X_{11}$:$X_{2010}$) were set to the value one. At this stage, the initial configurations of the optimization model are complete, and the *Solver* is ready to present the model results at once.

Once the *Solver* is initiated, as shown in Figure 7, the total demand cells (N23:W23) will be satisfied and filled with the requested supply units (Cells, X3:X22) within the SPL

capacity limit. At this phase, the LP model increases the number of decision variables ($X_{ji}$) next to the most competitive SPLs and decreases it once the SPL is insufficient. For example, the model converted all decision variables (from $X_{11}$ to $X_{201}$) into zero. Exceptionally, $X_{101}$ was set to 25,667 SLTs to satisfy the demand of the first WST $P_1$. This exception means all SPLs beneath WST $P_1$ (except $S_{10}$) were insufficient to supply that WST with the most competitive material logistics cost. The same procedure was performed with the rest of the WSTs (from $P_2$ to $P_{10}$). After these calculations, the LP model returns the objective function (Cell Z25) value that represents a product of decision variables ($X_{ji}$) and discount costs ($C_{ji}$) of the selected competitive material SPLs chosen above. The value of the objective function equals the minimum optimized total cost of all materials needed for the entire project based on competitive SPLs.

Accordingly, by utilizing dynamic and high accurate data, the outcomes of the proposed DC framework could assist the CSMs in saving logistics costs. As a sign of its reliability, the LP model used in this framework focused on specific SPLs and dynamically ignored the other. In other words, the model selected the material SPL $S_{10}$ as the best source for WSTs $P_1$ to $P_3$ due to its total price and capacity at optimization runtime. Similarly, the model decided to cover the material demand of $P_4$ exclusively from $S_{16}$. However, $P_5$ has a dual relationship with $S_{10}$ and $S_{19}$ to meet its high demand for crushed stone.

On the other hand, $S_{19}$ remained the best source for many other WSTs. It could completely cover the demand for roadworks from $P_6$ to $P_{10}$. Nevertheless, the alternatives mentioned above are variable and could be altered rapidly due to any slight change in the framework's primary input data, which IoT reflects. In the last module of section B, the framework summarizes all essential information associated with the competitive SPLs in a report. Thus, CSMs can select among many cost choices listed in the report to gain the desired limestone based on reliable and dynamic data sources with less human intervention.

In the real-life application of this model, the *Solver* can repeatedly calculate the minimum optimized cost of demand "individually" for the current WST, where the active CSM positions itself. This reliable logistics cost cannot be achieved without the DC concept utilized in this paper. Otherwise, the results of the logistics will be non-competitive due to the inaccurate and outdated supplier data. Moreover, connecting the CSM permanently with the OD over the IoT network protocols can provide the project with a variable list of competitive material suppliers due to the dynamic data shared through this framework. Thus, this framework contributes to better and more accurate cost optimization and DC for dynamic supplier selection.

### 4.4. Validation of the Smart Framework

In this section, the framework is validated by using two means. The first means compares the optimized model with the manually calculated cost to validate the reliability of the framework outcomes. The second means is comparing the optimized model results with three local SPLs to demonstrate the reasonability of the model results.

In the first means, the authors considered two results to prove the reliability of the proposed model. First, the optimized results were obtained from the LP model (Figure 7). Second, a list of manually calculated unit discount costs $C_{ji}$ (Table 4) represents the traditional plan for all project nodes. We can compare optimal vs. traditional outcomes with these results and decide whether the model is a reliable choice. It is to be noted that the optimal plan is the best distribution plan, which cannot be improved further within the given constraints. As the cost of material changed among all SPLs in real life, the little cost in the material purchasing process is due to the project team procuring more quantity from the supplier, which charges a lower price than other sources but costs more on transportation. Therefore, it was expected to save cost if the framework's optimal plan was fed by the CSMs instead of the manually prepared traditional plan. The deficits of the traditional plan consider one supplier each WST and neglect the newly updated cost alternatives revised permanently by the suppliers.

| | M | N | O | P | Q | R | S | T | U | V | W | X | Y | Z |
|---|---|---|---|---|---|---|---|---|---|---|---|---|---|---|
| 2 | Si | P1 | P2 | P3 | P4 | P5 | P6 | P7 | P8 | P9 | P10 | Total Supply | | *Constraints (Trucks)* |
| 3 | S1 | 0 | 0 | 0 | 0 | 0 | 0 | 0 | 0 | 0 | 0 | 0 | ≤ | 66,000 |
| 4 | S2 | 0 | 0 | 0 | 0 | 0 | 0 | 0 | 0 | 0 | 0 | 0 | ≤ | 46,200 |
| 5 | S3 | 0 | 0 | 0 | 0 | 0 | 0 | 0 | 0 | 0 | 0 | 0 | ≤ | 75,900 |
| 6 | S4 | 0 | 0 | 0 | 0 | 0 | 0 | 0 | 0 | 0 | 0 | 0 | ≤ | 23,100 |
| 7 | S5 | 0 | 0 | 0 | 0 | 0 | 0 | 0 | 0 | 0 | 0 | 0 | ≤ | 132,000 |
| 8 | S6 | 0 | 0 | 0 | 0 | 0 | 0 | 0 | 0 | 0 | 0 | 0 | ≤ | 59,400 |
| 9 | S7 | 0 | 0 | 0 | 0 | 0 | 0 | 0 | 0 | 0 | 0 | 0 | ≤ | 62,700 |
| 10 | S8 | 0 | 0 | 0 | 0 | 0 | 0 | 0 | 0 | 0 | 0 | 0 | ≤ | 50,820 |
| 11 | S9 | 0 | 0 | 0 | 0 | 0 | 0 | 0 | 0 | 0 | 0 | 0 | ≤ | 102,300 |
| 12 | S10 | 25,667 | 10,267 | 12,833 | 0 | 0 | 0 | 0 | 0 | 0 | 0 | 48,767 | ≤ | 112,200 |
| 13 | S11 | 0 | 0 | 0 | 0 | 0 | 0 | 0 | 0 | 0 | 0 | 0 | ≤ | 198,000 |
| 14 | S12 | 0 | 0 | 0 | 0 | 0 | 0 | 0 | 0 | 0 | 0 | 0 | ≤ | 118,800 |
| 15 | S13 | 0 | 0 | 0 | 0 | 0 | 0 | 0 | 0 | 0 | 0 | 0 | ≤ | 95,700 |
| 16 | S14 | 0 | 0 | 0 | 0 | 0 | 0 | 0 | 0 | 0 | 0 | 0 | ≤ | 138,600 |
| 17 | S15 | 0 | 0 | 0 | 0 | 0 | 0 | 0 | 0 | 0 | 0 | 0 | ≤ | 66,000 |
| 18 | S16 | 0 | 0 | 0 | 20,533 | 25,667 | 11,367 | 0 | 0 | 0 | 0 | 57,567 | ≤ | 89,100 |
| 19 | S17 | 0 | 0 | 0 | 0 | 0 | 0 | 0 | 0 | 0 | 0 | 0 | ≤ | 105,600 |
| 20 | S18 | 0 | 0 | 0 | 0 | 0 | 0 | 0 | 0 | 0 | 0 | 0 | ≤ | 128,700 |
| 21 | S19 | 0 | 0 | 0 | 0 | 0 | 4,033 | 17,967 | 10,267 | 7700 | 12,833 | 52,800 | ≤ | 52,800 |
| 22 | S20 | 0 | 0 | 0 | 0 | 0 | 0 | 0 | 0 | 0 | 0 | 0 | ≤ | 33,000 |
| 23 | Total Demand | 25,667 | 10,267 | 12,833 | 20,533 | 25,667 | 15,400 | 17,967 | 10,267 | 7700 | 12,833 | 159,133 | | |
| 24 | | = | = | = | = | = | = | = | = | = | = | | | Objective Fuction |
| 25 | Demand | 25,667 | 10,267 | 12,833 | 20,533 | 25,667 | 15,400 | 17,967 | 10,267 | 7700 | 12,833 | Total Cost | | 163,381,057 |
| 26 | | | | | | | | | | | | | | Goal: Min |

**Figure 7.** Optimal solutions calculated by *MS Excel Solver*.

Although supplier ($S_{19}$) has limited capacity, the *Solver* filled the demand of workstation $P_5$ "partially" from supplier $S_{19}$ due to its' competitive material price. Nevertheless, the *Solver* decided to cover the rest of the demand of $P_5$ from another source, the second priority pricing supplier, namely $S_{16}$. This step occurs automatically in our optimal plan, namely the smart framework, while the cost could be higher in traditional plans for the same WST.

**Table 4.** Unit discount costs $C_{ji}$ (measured by SLTs) calculated manually using the traditional plan.

| $S_i$ | $P_1$ | $P_2$ | $P_3$ | $P_4$ | $P_5$ | $P_6$ | $P_7$ | $P_8$ | $P_9$ | $P_{10}$ |
|---|---|---|---|---|---|---|---|---|---|---|
| $S_1$ | 2281 | 2069 | 2089 | 1853 | 1769 | 1917 | 2285 | 2537 | 2585 | 2765 |
| $S_2$ | 1050 | 1272 | 1609 | 1718 | 2024 | 2318 | 3041 | 3097 | 3292 | 3440 |
| $S_3$ | 908 | 1088 | 1407 | 1658 | 1909 | 2152 | 2670 | 2838 | 2878 | 2997 |
| $S_4$ | 720 | 1021 | 1553 | 1863 | 2448 | 2682 | 3609 | 3780 | 3951 | 4154 |
| $S_5$ | 742 | 890 | 1150 | 1309 | 1513 | 1711 | 2156 | 2239 | 2321 | 2420 |
| $S_6$ | 769 | 930 | 1213 | 1388 | 1612 | 1825 | 2320 | 2401 | 2500 | 2608 |
| $S_7$ | 888 | 1098 | 1426 | 1713 | 2003 | 2261 | 2818 | 2960 | 3064 | 3192 |
| $S_8$ | 704 | 958 | 1408 | 1784 | 2164 | 2480 | 3259 | 3418 | 3547 | 3718 |
| $S_9$ | 1196 | 1463 | 1943 | 2349 | 2734 | 3099 | 3964 | 4120 | 4271 | 4456 |
| $S_{10}$ | 695 | 838 | 1098 | 1316 | 1536 | 1732 | 2209 | 2262 | 2346 | 2443 |
| $S_{11}$ | 1056 | 1197 | 1459 | 1681 | 1879 | 2075 | 2556 | 2622 | 2721 | 2840 |
| $S_{12}$ | 992 | 1146 | 1374 | 1562 | 1752 | 1922 | 2334 | 2379 | 2452 | 2535 |
| $S_{13}$ | 919 | 1048 | 1282 | 1480 | 1702 | 1848 | 2268 | 2344 | 2420 | 2510 |
| $S_{14}$ | 1149 | 1363 | 1749 | 2076 | 2386 | 2680 | 3376 | 3501 | 3627 | 3772 |
| $S_{15}$ | 1065 | 1227 | 1665 | 2031 | 2375 | 2704 | 3537 | 3500 | 3840 | 4007 |
| $S_{16}$ | 2443 | 2126 | 1607 | 1211 | 990 | 1273 | 2206 | 2184 | 2580 | 3064 |
| $S_{17}$ | 1024 | 1112 | 1409 | 1654 | 1907 | 2112 | 2672 | 2647 | 2887 | 2999 |
| $S_{18}$ | 1043 | 1229 | 1724 | 2140 | 2531 | 2905 | 3845 | 3808 | 4194 | 4383 |
| $S_{19}$ | 1885 | 1743 | 1515 | 1337 | 1273 | 997 | 959 | 1007 | 1131 | 1387 |
| $S_{20}$ | 1207 | 1239 | 1571 | 2107 | 2367 | 2727 | 3591 | 3551 | 3943 | 4255 |

In the second means, three reference SPLs were interviewed to seek their opinion about the unit price of the material logistics cost. In our case, the optimal total cost between the two cities was around SAR 163 MM. The road distance of the project was 700 km; thus, the optimal unit price for transported material is [SAR 163,381,057 $\div$ (700,000 m $\cdot$ 0.5 m $\cdot$ 22 m)] = 21.22 SAR/m$^3$ based on the optimal total cost calculated by the LP model. On the other hand, the first reference, SPL (A), stated a range of limestone unit prices between SAR 20 and SAR 25, excluding transportation costs. The second reference, SPL (B), stated that the unit price of this type of road material equals SAR 40 excluding transportation as well. However, this price includes the overhead cost and profit (mark-up). Of note, the mark-up is usually 40% of the price in Saudi Arabia. The third SPL (C) implies a competitive range of material unit prices between SAR 18 and 26, excluding transportation. Having this in mind, Table 5 illustrates a comprehensive comparison of the average cost given by the reference SPLs. The cost covers shipping one unit of limestone, including transportation cost, excluding overhead cost. Additionally, the last column contains the optimal unit prices advanced earlier in this study for comparison purposes.

**Table 5.** The optimal unit discount cost $C_{ji}$ vs. the cost gained from reference SPLs.

| Cost Type | SPL (A) | SPL (B) | SPL (C) | Optimal Unit Cost |
|---|---|---|---|---|
| | SAR/m$^3$ | SAR/m$^3$ | SAR/m$^3$ | SAR/m$^3$ |
| Average material price (excl. overhead cost) | 22.5 | 24 | 22 | 17.22 |
| Transported (+SAR 4.0) | 26.5 | 28 | 26 | 21.22 |

Note that the average cost of transporting one cubic meter of material ranges between SAR 3.0 and SAR 5.0 per kilometer in KSA. For calculation simplicity, the average transportation cost is considered SAR 4.0 for further usage in the following table. All reference supplier names are omitted in this paper due to privacy rights.

Overall, the optimal cost reached in this study is considered competitive, even with transportation costs. It outperforms the closest competitor SPL C, saving SAR 4.78 (26 − 21.22) of cost for each cubic meter transported one kilometer. Considering the entire project length and the difference of cost, the potential saving exceeds SAR 36.8 MM in favor of the optimal solution, as shown in the following calculation:

$$\text{Potential saving} = \text{road length} \times \text{width} \times \text{average depth} \times \text{difference}$$
$$= 700,000 \text{ m} \times 22 \text{ m} \times 0.50 \text{ m} \times 4.78 \text{ SAR/m}^3 \tag{6}$$
$$= \text{SAR } 36,806,000$$

The saving for each WST could be higher if the accurate GPS coordinates are considered continuously through this framework. However, such calculation needs powerful software. Such colossal research can be performed in future studies.

*4.5. Sensitivity Analysis*

Supplier selection could be single sourcing (one SPL) or multiple sourcing (group of SPLs) is/are selected to fulfill the entire project's demand. Randomly biding suppliers can lead to a massive economic loss when the whole project demand is taken from a single source. This section demonstrates the saving cost by comparing the results of single and multiple sourcing. For example, the total cost of material shipped from a single source ($S_1$) to the entire project exceeded SAR 342 MM. Similarly, the cost amounted to SAR 303 MM when the calculation depended only on ($S_{11}$). This type of calculation is usually used in traditional plans. The results of these costly attempts are presented in the following scenario, Figures 8 and 9:

| Si | P1 | P2 | P3 | P4 | P5 | P6 | P7 | P8 | P9 | P10 | Total Supply | | Constraints (Trucks) |
|---|---|---|---|---|---|---|---|---|---|---|---|---|---|
| S1 | 25,667 | 10,267 | 12,833 | 20,533 | 25,667 | 15,400 | 17,967 | 10,267 | 7700 | 12,833 | 159,133 | ≤ | 200,000 |
| Total Demand | 25,667 | 10,267 | 12,833 | 20,533 | 25,667 | 15,400 | 17,967 | 10,267 | 7700 | 12,833 | 159,133 | | |
| | = | = | = | = | = | = | = | = | = | = | | | **Objective Fuction** |
| Demand | 25,667 | 10,267 | 12,833 | 20,533 | 25,667 | 15,400 | 17,967 | 10,267 | 7700 | 12,833 | Total Cost | | 342,059,667 |
| | | | | | | | | | | | | | Goal: Min |

**Figure 8.** Scenario1 for SPL $S_1$.

| Si | P1 | P2 | P3 | P4 | P5 | P6 | P7 | P8 | P9 | P10 | Total Supply | | Constraints (Trucks) |
|---|---|---|---|---|---|---|---|---|---|---|---|---|---|
| S11 | 25,667 | 10,267 | 12,833 | 20,533 | 25,667 | 15,400 | 17,967 | 10,267 | 7700 | 12,833 | 159,133 | ≤ | 200,000 |
| Total Demand | 25,667 | 10,267 | 12,833 | 20,533 | 25,667 | 15,400 | 17,967 | 10,267 | 7700 | 12,833 | 159,133 | | |
| | = | = | = | = | = | = | = | = | = | = | | | **Objective Fuction** |
| Demand | 25,667 | 10,267 | 12,833 | 20,533 | 25,667 | 15,400 | 17,967 | 10,267 | 7700 | 12,833 | Total Cost | | 303,037,401 |
| | | | | | | | | | | | | | Goal: Min |

**Figure 9.** Scenario2 for SPL $S_{11}$.

It is noteworthy that the optimized logistics cost of the entire road construction project is still favorable, with a cost less than SAR 163 MM (equivalent to around USD 43.5 MM) compared to Scenarios 1 and 2. The optimal solution could save more than 40% of the logistics cost than single-sourcing solutions. Moreover, the savings can be higher with DC if the calculation is based on moving WST coordinates, as demonstrated earlier.

## 5. Conclusions

Road projects are usually fraught with many challenges, including material logistics costs. They often rely on inaccurate SPL data, which makes it difficult to determine the cost, quantity, quality, and transportation duration of the demanded project materials. The wrong choice of material suppliers can lead the supply chain to suffer losses, directly affecting the project's performance. In this regard, many studies conducted material logistics optimization models for road projects; however, the majority based their decisions on inaccurate or outdated data. Hence, there is a need to enhance supplier selection models to optimize material logistics costs with updated and validated data.

This paper contributes a framework that utilizes many smart elements to feed selection and optimization models with accurate, dynamic, and reliable material data to fix this gap. Besides the LP model, this framework utilizes the fourth industry revolution components such as IoT technologies, OD, BC, MSW, and CSM. Only quantitative criteria were considered as input data to the LP optimization model to select suppliers and calculate optimal costs. A case study was used to prove the framework's reliability. The case study is a planned road construction between two Saudi cities, including building a subbase course of a highway road for a 700 km stretch with 10 workstations and 20 suppliers, considering the moving GPS coordinates of the workstations. The framework demonstrated high cost-saving levels of the entire project.

Through IoT network protocols and with less human intervention, CSM fed the framework with input data about the demand of the workstations. In response, the framework fetched and validated the supplier's data from OD and calculated the material logistics cost for each supplier. Using *MS Excel Solver*, the LP model delivered an optimized material logistics cost and selected the competitive suppliers for the individual workstations, including the entire project considering the GPS coordinates of the permanently moving roadworks.

The framework was validated using two means—the first one, comparing the optimized results with the traditionally calculated costs. The second means comparing the optimized results with the cost gained from three reference suppliers. The first comparison results revealed that the traditional plan considers a single supplier for each workstation

and neglects the updated cost data revised by the suppliers. However, the proposed framework in this paper was sensitive for any data updates and could change the optimal plans in response.

On the other hand, three cost alternatives were calculated for the entire project based on the unit price data gained from three reference suppliers. The results were compared with the frameworks' optimal outcomes. The optimal cost outperformed the closest competitor with a potential saving of SAR 36.8 MM. However, the cost-saving for each WST can be even higher if the accurate GPS coordinates are considered continuously through this framework. The sensitivity analysis demonstrated that the cost-saving could exceed 40% of the entire project costs by using multiple instead of single sourcing. This framework utilized IoT technologies and enhanced the supplier selection process with dynamic data sources for more material logistics cost calculations accuracy.

It is recommended in this study to develop the model with more constraints and parameters, such as various SLT sizes; internal transportation among project sites; moving WST coordinates; considering traffic limitation of trucks in real-life; using multiple types of construction materials; quality; delivery performance; cost; and capability. However, the price remains the primary factor of the selection process. Future research of this work can be a fully coded program of the integrated proposed model in this paper to attain an entirely automated process. In addition, this model can be employed to optimize any material logistics associated with the project.

**Author Contributions:** Conceptualization, A.A. (Abdulkareem Alanazi) and K.A.-G.; Data curation, A.A. (Abdulkareem Alanazi); Formal analysis, A.A. (Abdulkareem Alanazi); Funding acquisition, K.A.-G. and A.A. (Abdullah Alsugair); Investigation, A.A. (Abdulkareem Alanazi) and K.A.-G.; Methodology, A.A. (Abdulkareem Alanazi) and K.A.-G.; Project administration, K.A.-G. and A.A. (Abdullah Alsugair); Resources, K.A.-G.; Software, A.A. (Abdulkareem Alanazi); Supervision, K.A.-G. and A.A. (Abdullah Alsugair); Validation, A.A. (Abdulkareem Alanazi) and K.A.-G.; Visualization, K.A.-G. and A.A. (Abdullah Alsugair); Writing—original draft, A.A. (Abdulkareem Alanazi); Writing—review and editing, A.A. (Abdulkareem Alanazi), K.A.-G. and A.A. (Abdullah Alsugair) All authors have read and agreed to the published version of the manuscript.

**Funding:** This research was funded by the Deanship of Scientific Research, King Saud University.

**Institutional Review Board Statement:** Not applicable.

**Informed Consent Statement:** Not applicable.

**Data Availability Statement:** The raw data supporting the findings of this paper are available on request from the corre-sponding author.

**Acknowledgments:** The authors thank the Deanship of Scientific Research, King Saud University, for funding and supporting this research.

**Conflicts of Interest:** The authors declare no conflict of interest, financial or otherwise.

## Abbreviations

| | |
|---|---|
| IoT | internet of things |
| DC | data connectivity |
| BC | blockchain |
| AI | artificial intelligence |
| KSA | Kingdom of Saudi Arabia |
| LP | linear programming |
| GPS | global positioning system |
| MIM | ministry of industry and mineral resources |

| OD | governmental open data |
|---|---|
| MSW | mapping software |
| CSM | construction smart machine |
| WST | workstation |
| SPL | supplier |
| SAR | Saudi Arabian Riyal (currency) |
| MM | million |
| j | index of workstations |
| i | index of suppliers |
| $P_j$ | symbol of workstation |
| $S_i$ | symbol of supplier |
| $X_{ji}$ | decision variable of unit quantity for shipping material from SPL to WST |
| $C_{ji}$ | variable of unit discount cost for shipped material. |
| $D_{ji}$ | distance variable measured by MSW to ship material from SPL to WST. |
| $T_j$ | unit demand of material (loaded SLT) needed for WST j |
| $R_i$ | transportation price per km and SLT |
| $M_i$ | dynamically updated material prices offered by SPL. |
| $V$ | SLT volume |
| $\rho_i$ | maximum supply capacity of raw material available by SPL |
| $\mu_j$ | total demand of raw material (loaded SLTs) needed for WST j |
| km | kilometer |

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
