# Peer review of "Framework for Smart Cost Optimization of Material Logistics in Construction Road Projects"

_infrastructures, doi:10.3390/infrastructures7050062_

Round 1

Reviewer 1 Report

I don’t see any reason for initial capital letters for such words: Smart Construction, Linear Programming, etc. The proportion of old sources is extremely high.  The manuscript has a low integrative value in the current research. material from a supplier (SPL) – what is SPL? Figure 1 is unclear and not uniformly designed. A large proportion of the manuscript needs to be substantially rewritten to provide a more robust analysis of the issues that it raises. The manuscript will benefit from further discussion of key concepts and methodological criteria in order to offer a better articulation between theory and data. Some bibliographic references are simply brought up without being developed, or without an adequate explanation as to why they are relevant. There is a need of structuring the discussion to ensure that the methodological aspects are clearly presented. Many statements appear in the discussion section without explanation as to the data on which they are based. Several statements made in the paper are not supported by adequate empirical evidence or by making reference to relevant literature. The conclusion should clarify the main contribution of the paper and the value added to the field.

The relationship between sustainable urban governance networks and Internet of Things-based real-time production logistics as regards construction road projects has not been covered, and thus such recent sources should be cited:

Evans, V., and Horak, J. (2021). “Sustainable Urban Governance Networks, Data-driven Internet of Things Systems, and Wireless Sensor-based Applications in Smart City Logistics,” Geopolitics, History, and International Relations 13(2): 65–78. doi: 10.22381/GHIR13220215.

Popescu, G. H., Petreanu, S., Alexandru, B., and Corpodean, H. (2021). “Internet of Things-based Real-Time Production Logistics, Cyber-Physical Process Monitoring Systems, and Industrial Artificial Intelligence in Sustainable Smart Manufacturing,” Journal of Self-Governance and Management Economics 9(2): 52–62. doi: 10.22381/jsme9220215.

Nica, E., Stan, C. I., Luțan (Petre), A. G., and Oașa (Geambazi), R.-Ș. (2021). “Internet of Things-based Real-Time Production Logistics, Sustainable Industrial Value Creation, and Artificial Intelligence-driven Big Data Analytics in Cyber-Physical Smart Manufacturing Systems,” Economics, Management, and Financial Markets 16(1): 52–62. doi: 10.22381/emfm16120215.

Grant, D.B., Dabija, D.C., Tumoala, V., 2021. Reshaping Green Retail Supply Chains in a New World Order. In: Taylor, A. (Ed.). Rethinking Leadership for a Green World. Routledge, 282-309.

Author Response

We would like to thank the reviewer for his/her valuable comments to improve the paper. We did our best to answer the comments. Please see the attachment file for answering each comment.

Reviewer 2 Report

Overall the field of the paper is topical and becomes even more actual in modern rapidly changing world.

However reviewer did not fully understand the contribution and the message of the paper.

From one hand authors propose "dynamic and optimized framework for  material logistics", perform deep literature review on related topics such as IoT, blockchain, optimization methods and criteria. 

But from the other hand the contribution part of the paper describes application of MS Excel for solving the transportation LP problem, which is well known example for Solver.

The use case in general is very impressive. However from reviewer perspective the demonstration of the proposed framework is limited to resolving transportation LP problem repeatedly several times in MS Excel.

For the reviewer it is not clear how all these dynamic values from smart IoT devices appear in the Excel tables and how the results of optimization are further used in dynamic way? The paper lacks these details, especially if it aims to present the framework for such tasks.

In terms of dynamic nature of the transportation problems, it is not clear how the framework will handle:

  • unexpected supply shortages;
  • unexpected demand;
  • mid-process interruptions in delivery, workstations, mechanisms, etc;
  • multiple criteria (not only costs);
  • time domain for planning.

Few more remarks:

  • Section 2.3 has title "supplier selection criteria", but the actual content is about selection methods.
  • Section 2.4 discusses only about LP approaches.
  • The paper title includes "optimized framework", which means at least some alternative frameworks should be considered and evaluated. Perhaps better title would be "the framework for optimization".

Author Response

(The authors gave the same response as above.)

Round 2

Reviewer 1 Report

The revised version can be published.

Author Response

The authors would like to thank the reviewer for his/her valuable comments to improve the paper. Much appreciation

Reviewer 2 Report

Reviewer still did not get an answer for this question: "how all these dynamic values from smart IoT devices appear in the Excel tables and how the results of optimization are further used in dynamic way?"

There are named IoT modules, values, but how technically it is implemented is not clear yet.

Author Response

The authors would like to thank the reviewer for his/her valuable comments to improve the paper. We are much appreciated. We try our best to answer the reviewer's comments to the best of our understanding. Please refer to our attached comments answer file.
